Use of integrated population models for assessing density-dependence and juvenile survival in Northern Bobwhites (Colinus virginianus)

http://orcid.org/0000-0003-4978-6027 Lewis William B. 1 wblewis7@gmail.com
Nater Chloé R. 2
Rectenwald Justin A. 3
Sisson D. Clay 3
Martin James A. 1
1 Warnell School of Forestry and Natural Resources, University of Georgia , Athens, Georgia , United States
2 Norwegian Institute for Nature Research , Trondheim , Norway
3 Albany Quail Project, Tall Timbers Research Station , Tallahassee, Florida , United States
Caravaggi Anthony
Electronic publication date: 2024 Dec 4
Publication date: 2024
Volume: 12
Electronic Location ID: e18625
Received 2024 Aug 13; Accepted 2024 Nov 11
Copyright: © 2024 Lewis et al.
Copyright year: 2024
Copyright holder: Lewis et al.
License: This is an open access article distributed under the terms of the Creative Commons Attribution License, which permits unrestricted use, distribution, reproduction and adaptation in any medium and for any purpose provided that it is properly attributed. For attribution, the original author(s), title, publication source (PeerJ) and either DOI or URL of the article must be cited.
License URL: https://creativecommons.org/licenses/by/4.0/

Keywords: Integrated population models, Northern bobwhite, Density-dependence, Demography, Per-capita productivity, Abundance estimation, Juvenile survival, Population dynamics

Funding: Warnell School of Forestry and Natural Resources University of Georgia Albany Quail Project/Tall Timbers Funding was provided by the Warnell School of Forestry and Natural Resources, University of Georgia, and by the Albany Quail Project/Tall Timbers. The funders had no role in study design, data collection and analysis, decision to publish, or preparation of the manuscript.

==============================
Management of wildlife populations is most effective with a thorough understanding of the interplay among vital rates, population growth, and density-dependent feedback; however, measuring all relevant vital rates and assessing density-dependence can prove challenging. Integrated population models have been proposed as a method to address these issues, as they allow for direct modeling of density-dependent pathways and inference on parameters without direct data. We developed integrated population models from a 25-year demography dataset of Northern Bobwhites (Colinus virginianus) from southern Georgia, USA, to assess the demographic drivers of population growth rates and to estimate the strength of multiple density-dependent processes simultaneously. Furthermore, we utilize a novel approach combining breeding productivity and post-breeding abundance and age-and-sex ratio data to infer juvenile survival. Population abundance was relatively stable for the first 14 years of the study but began growing after 2012, showing that bobwhite populations may be stable or exhibit positive population growth in areas of intensive management. Variation in breeding and non-breeding survival drove changes in population growth in a few years; however, population growth rates were most affected by productivity across the entire study duration. A similar pattern was observed for density-dependence, with relatively stronger negative effects of density on productivity than on survival. Our novel modeling approach required an informative prior but was successful at updating the prior distribution for juvenile survival. Our results show that integrated population models provide an attractive and flexible method for directly modeling all relevant density-dependent processes and for combining breeding and post-breeding data to estimate juvenile survival in the absence of direct data.

Introduction

Effectively managing wildlife populations necessitates accurate estimates of vital rates and an understanding of how vital rates affect population growth rates (Caswell, 2001; Schaub & Abadi, 2011; Mills, 2012; Zipkin & Saunders, 2018). Importantly, wildlife managers must understand the demographic drivers of population change to target management actions towards appropriate vital rates which will most cost-effectively promote or maintain managed populations (Caswell, 2000; Wisdom, Mills & Doak, 2000). Population management also requires accurate estimates of the strength of density-dependent feedback on vital rates. Understanding density-dependent effects on population growth is critically important for understanding population regulation and projecting population trajectories and extinction risk (Ginzburg, Ferson & Akçakaya, 1990; Hixon, Pacala & Sandin, 2002; Sæther et al., 2004). Density-dependence is also one prerequisite for compensatory mortality, so estimating the strength and effects of density-dependence is required for setting informed harvest regulations (Lebreton, 2005; Péron, 2013). Not accounting for density-dependent processes in conservation planning may limit the effectiveness of management actions, potentially leading to a waste of conservation resources (Pöysä & Pöysä, 2002; Guthery & Shaw, 2013).

Identifying density-dependent processes in wildlife populations can prove challenging in practice, as these effects generally require long time-series data to detect (Lebreton, 2009; Brook & Bradshaw, 2006; Knape & de Valpine, 2012; Guthery & Shaw, 2013). Furthermore, population density may affect multiple demographic rates simultaneously or only influence life stages at certain times during the annual cycle or under specific environmental circumstances (Ratikainen et al., 2008; Lebreton & Gimenez, 2013). Many studies testing for density-dependence have used correlation analyses to compare time series of vital or population growth rates to variation in some metric of abundance (e.g., Fryxell et al., 1999; Sillett & Holmes, 2005; Tavecchia et al., 2007). While simple to perform, these methods suffer from several limitations which can limit power to detect density-dependence or lead to erroneous conclusions. First, these analyses separately estimate correlations between density and each vital rate instead of with a unified framework (Abadi et al., 2012). Moreover, these methods often relate observed demographic rates to count data; however, observed counts may be biased relative to the true population parameters due to observation error (Lebreton, 2009; Abadi et al., 2012; Lebreton & Gimenez, 2013). State-space models have been used to account for observation error when assessing density-dependence, though parameters may often be difficult to estimate (Lebreton & Gimenez, 2013).

Integrated population models (IPMs) provide an ideal framework for cohesively modeling density-dependent effects on multiple demographic parameters at the same time (Abadi et al., 2012). IPMs combine data from multiple independent data sources (e.g., trapping data, surveys, radiotelemetry) into a single unified model of population dynamics by maximizing the joint likelihood (Schaub & Abadi, 2011; Zipkin & Saunders, 2018). By incorporating multiple datasets into a single model, IPMs can properly propagate uncertainty from all datasets and can potentially compensate for sparse datasets or those showing conflicting patterns (Schaub & Abadi, 2011; Saunders, Cuthbert & Zipkin, 2018; Saunders et al., 2019). IPMs also allow for direct assessment of density-dependent effects on population growth. These models can separately estimate the detection and ecological state processes and can directly model the effects of population abundance on multiple demographic rates simultaneously, though incorporating density-dependent effects often increases computation time (Abadi et al., 2012). Because they incorporate all relevant data into a single model, IPMs often allow for more accurate and precise estimates of demographic rates and have a higher power for detecting density-dependence compared to analyzing datasets independently (Besbeas et al., 2003; Abadi et al., 2012; Eacker et al., 2017).

Due to explicitly modeling all of the demographic processes affecting population dynamics, IPMs also allow for estimation of parameters which may be difficult or impossible to estimate from separate analyses (Schaub et al., 2007; Abadi et al., 2010a, 2010b; Plard et al., 2019). One such parameter is juvenile survival. Variation in juvenile survival can be a key driver of population growth (Fefferman & Reed, 2006; Sinnott et al., 2023); however, this parameter is often difficult to assess, and so data is lacking for many wildlife populations. Even without direct survival data, this parameter could be inferred from an IPM framework. Our method of inferring juvenile survival is based on Arnold (2018), who used the ratio of juvenile to adult birds captured in the fall to estimate the previous breeding season’s fecundity. This estimate of fecundity is a combined measure of chick production and juvenile survival to the capture period. Juvenile survival cannot be separated from productivity in the framework of Arnold (2018); however, this parameter could be estimated if independent productivity data are available. An IPM framework can be used to estimate the number of breeding adults in the population, which can then be combined with estimates of per-capita productivity rates from breeding productivity data to estimate the total number of young produced during the breeding season. Similarly, the number of young surviving to the post-breeding season can be estimated by combining post-breeding surveys (to estimate post-breeding population sizes) and age ratios from trapping or harvest data. The difference between the number of young produced during the breeding season and the number of young surviving to post-breeding can then be used to infer rates of juvenile survival. Though inferred estimates may be imprecise or biased (Plard et al., 2019; Riecke et al., 2019), IPMs can be used to estimate juvenile survival in the absence of direct data by incorporating nest monitoring, fall counts, and age ratio data.

We developed an IPM to assess population dynamics and estimate density-dependent effects and juvenile survival in a population of Northern Bobwhites (Colinus virginianus, hereafter, bobwhites) in southern Georgia, USA. Bobwhite populations have been declining across much of the geographic range over at least the past 50 years (Hernández et al., 2013; Janke et al., 2017; Sauer, Link & Hines, 2020), though local populations may be stable or increasing in areas of intensive management (McConnell et al., 2018). Declines are often attributed to habitat loss and landscape change, but other threats, such as diseases, changes in predation pressure, and climate change, have also been implicated (Rollins & Carroll, 2001; Hernández et al., 2013; Brennan, Hernandez & Williford, 2020).

Bobwhites are one of the most intensively studied species in North America due to their high cultural value. Despite a history of bobwhite research shaping early notions of the importance of density-dependence for population regulation (Errington, 1941, 1945), previous bobwhite population models have either ignored density-dependent processes (Sandercock et al., 2008; Rosenblatt et al., 2021; Sinnott et al., 2023) or only incorporated effects on recruitment (McConnell et al., 2018). Previous work has suggested that density-dependent effects on overwinter survival may be important for population persistence (DeMaso et al., 2013); however, no studies have incorporated density-dependent effects on both the recruitment and survival processes in bobwhite population models. Previous bobwhite IPMs have also been unable to generate precise estimates of juvenile survival without incorporating direct survival data (Rosenblatt et al., 2021; Sinnott et al., 2023). Juvenile survival data are often difficult and costly to collect and may result in biased estimates. For example, previous studies have directly assessed juvenile survival of bobwhites through methods such as mark-recovery and suturing radiotransmitters onto chicks (Terhune, Chandler & Martin, 2017; Terhune, Palmer & Wellendorf, 2019). Mark-recovery requires large sample sizes while capture stress and the extra weight of bands and/or transmitters could artificially increase mortality rates for this already-vulnerable life history stage (Trefry, Diamond & Jesson, 2013), necessitating alternative methods for estimating bobwhite juvenile survival. The primary feathers of juvenile bobwhites grow at a predictable pace during the first 5 months of their life, allowing reliable determination of the hatch month of young birds captured or harvested in the fall (Rosene, 1969). Integrated population models could therefore estimate bobwhite juvenile survival rates separately for each monthly cohort by subdividing productivity and post-breeding age-ratio data by each month of the breeding season. While population models generally incorporate constant juvenile survival within a given year, estimating a separate survival rate for birds hatched in each month of the breeding season would account for variation in juvenile survival due to factors such as seasonal changes in environmental conditions (Grüebler & Naef-Daenzer, 2008; Terhune, Palmer & Wellendorf, 2019; White et al., 2021). The objectives of our study were threefold. First, we assessed the demographic drivers of population growth rates in the study population of bobwhites to inform future management actions. Second, we directly incorporated density-dependent effects on multiple demographic rates to assess their relative effects on population dynamics. Finally, we assessed the feasibility of incorporating breeding and post-breeding data to estimate monthly-cohort-specific juvenile survival in an IPM framework.

Materials and Methods

Study area

Our study site consisted of a 6,000-ha privately-owned property in Baker County within the Upper Coastal Plain physiographic region of southwest Georgia. The site consists largely of upland pine forests, predominantly of longleaf (Pinus palustris), slash (Pinus elliottii), and loblolly (Pinus taeda) pines, with scattered hammocks of southern live oak (Quercus virginiana) (Rectenwald et al., 2021). This property has been intensively managed since the 1940s to promote bobwhite populations and maintain appropriate habitat of low-density pine stands (3–9 m2/ha basal area) with understory early-successional vegetation dominated by broomsedge (Andropogon virginicus), blackberry (Rubus spp.), goldenrod (Solidago spp.), partridge pea (Chamaecrista fasciculata), and ragweed (Ambrosia artemisiifolia) (Terhune et al., 2006; Sisson et al., 2009). Approximately 20% of the area is occupied by scattered 1–4 ha fallow fields dominated by ragweed and partridge pea. Management actions undertaken at the site include timber thinning, herbicide application and mechanical removal to control hardwood encroachment, annual disking of fallow fields, mammalian predator trapping, year-round supplemental feeding, and prescribed burns (Rectenwald et al., 2021). Pine stands were burned on a 2-year rotation, with roughly half of forested areas burned each year (Rectenwald et al., 2021).

Data collection

Demographic and population data were collected at the study site from 1998–2022. We incorporated seven demographic datasets in our population model. Data collection was approved by the Tall Timbers Research Station (IACUC: TT-2024-01).

Datasets one and two: November trapping data

Bobwhites were trapped annually in the late fall (mid-October to mid-November) and spring (March through early April) using baited wire funnel traps (Stoddard, 1931). These trapping periods approximately corresponded to 1 month into the non-breeding season (fall) and right before the start of the breeding season (spring, see below). Fall trapping was completed right before the start of the hunting season. Captured birds were banded with a uniquely-numbered aluminum band and were classified by sex and age based on plumage characteristics. Birds were aged as either subadult (hatched in the previous breeding season and recruiting into the population) or adult (in at least their second year). We used the fall trapping data (hereafter, November trapping data) to calculate the number of adult males, adult females, and subadults captured in each year (dataset one). We did not divide the capture ratio for subadults by sex as we could not determine the sex of 15% of November subadults. We assumed a 50/50 sex ratio among subadults. The age of November subadults was estimated on capture based on the pattern of primary feather molt as in Rosene (1969). Hatch date was then estimated by backdating from the capture date. Subadults were grouped into one of four monthly cohorts (June, July, August, or September, see below for more details) based on the month of estimated hatch (dataset two).

Datasets three and four: November and December harvest data

We used harvest data from the early winter to provide another estimate of the hatch month cohort ratios of subadults. Routine hunting of bobwhites occurred in the study area approximately once every 2 weeks during the hunting season (mid-November–February) (Terhune et al., 2007). Hunting and harvest were completely controlled on the private property of the study site, so harvest and band reporting rates should be at or near 100% (Terhune et al., 2007; Sisson et al., 2009). Harvested birds were aged and sexed, and the age of subadults was determined based on the progression of primary molt as described above. Subadults can be reliably aged based on primary molt up to 150 days old, at which point the juvenile molt is finished (Rosene, 1969). A bird hatched on July 1st would complete its molt on November 28th; therefore, subadults harvested before this date can be reliably classified into June, July, August, and September hatch cohorts. We calculated the yearly number of birds from each monthly cohort amongst birds harvested from the start of hunting to November 27th, which we hereafter refer to as the November harvest data (dataset three). Similarly, birds harvested between November 28th and December 28th could be reliably assigned to August and September hatch cohorts, though birds from the June and July cohorts could no longer be reliably distinguished. For birds harvested during this period, we calculated the yearly number of subadults from the August, September, and combined June/July cohorts (hereafter, December harvest data, dataset four). We did not incorporate birds harvested after December 28th, as the majority of harvested subadults had completed their molt by the beginning of January. November–December harvest occurred in every year of the study except 2018, when a late-season hurricane impacted the area.

Dataset five: radiotelemetry data

We monitored survival of bobwhites throughout the annual cycle using radiotelemetry. Upon capture (see above), a subset of birds weighing at least 132 g were fitted with a 6.3 g necklace-style radiotransmitter with activity switch (Holohil Systems, Ltd., Ontario, Canada). Previous work from the region has found negligible effects of transmitter deployment on bobwhite survival (Palmer & Wellendorf, 2007; Terhune et al., 2007; Wann et al., 2020). Transmitters were roughly evenly deployed across sex and age categories in the fall. Transmitters were predominantly deployed on females in the spring, as the primary goal was to track females during the breeding season to find nests (see below); however, some males in each year survived from the previous fall and were tracked throughout the breeding season (mean: 15.5, range: 3–41). Birds were tracked at least twice weekly using the homing method (White & Garrott, 1990) until either confirmed mortality or censure (e.g., radio failure). Censoring only occurred when technicians lost track of a radiotransmitter, when the transmitter clearly fell off the bird, or in a very small number of cases (<0.5% of tracked birds) when the bird died or was otherwise removed from the population at a later capture. Terhune et al. (2007) found bobwhite survival estimates at a nearby study site to be comparable between known-fate telemetry and capture-recapture data, suggesting that any biases associated with incorrect censoring (e.g., a predator also destroying the transmitter when killing a bird) are likely to be minimal. We used the telemetry data to create bi-weekly capture histories for each bird. Capture histories were structured based on age, sex, biological year (starting April 1st), and season (breeding or non-breeding, see below).

Dataset six: nest monitoring data

Bobwhite nests were found by following females via radiotelemetry to identical locations on consecutive tracking events during the breeding season. Each nest was checked daily until either it succeeded (at least one egg hatched) or failed. Most nests hatched from June–September; therefore, we calculated the total number of chicks produced from successful nests in June, July, August, and September of each year. A small number (4%) of nests hatched at the end of May; these were binned with the June cohort. Males contribute to annual productivity by incubating a proportion of nests (Suchy & Munkel, 1993; Burger et al., 1995; Sinnott et al., 2023); therefore, we calculated monthly chick production separately for each to sex to estimate sex-specific productivity rates. The total number of chicks produced from male-and-female-incubated nests in each breeding month were used as data in the model. Estimating monthly sex-specific per-capita productivity rates also required information about the number of adults from which the chicks were produced, which we calculated based on the number of females and males alive at the start of each month from the radiotelemetry data. In some cases, the adult attending a nest did not have an attached radiotransmitter (e.g., the nest was found opportunistically rather than through radiotelemetry). We added these birds to the monthly counts of the number of alive males and females. We note that this formulation of the productivity data incorporates data from all nesting attempts during the breeding season, including renests and double-broods.

Dataset seven: covey counts

Bobwhite coveys (non-breeding aggregations) were surveyed annually during the fall (mid-October–mid-November) via the quadrat-sampling method to estimate post-breeding bobwhite abundance (Wellendorf, Palmer & Bromley, 2004; Howell, Terhune & Martin, 2021). A series of twelve 500 × 500 m grids were established throughout the study site in areas deemed to be representative of bobwhite habitat and to contain all habitat resources required throughout the annual cycle. Grids were surveyed synchronously by four observers standing at the midpoint of each of the sides of the grid. Surveys were completed from 45 min before sunrise to sunrise, with observers recording the location of all coveys heard calling within the grid on aerial maps. Observers later compared observations to ensure that coveys were not counted more than once. Only 8–10 of the 12 grids were sampled each year. Bird dogs were used to flush coveys at a subset of grids in each year (mean: 8.3 grids, range: 2–10) to estimate average covey size.

Statistical modelling

We used an Integrated Population Model to jointly estimate seasonal abundance, survival, and productivity of the bobwhite population from the seven demographic datasets (Fig. 1). As counts were only conducted at a series of 12 grids, our model specifically estimates the abundance across these grid cells. The grid cells were chosen to be representative of bobwhite habitat across the study area, so we divided abundance estimates by the total grid area (300 ha) to estimate density at the study site. We note that bobwhites in this region generally use similar spatial areas throughout the annual cycle, unlike populations from some other parts of the range (e.g., Lohr et al., 2011; Brooke et al., 2015). The model incorporates two main components: 1) a state process consisting of an age-, sex-, and season-structured population model relating change in population size and structure to demographic rates, and 2) observation processes relating population dynamics to observed data. Bobwhite population data were collected throughout the entire year, which we divided into the breeding season (April–September) and non-breeding season (October–March) (Sisson et al., 2009; Rosenblatt et al., 2021). We modeled the biological year as starting at the beginning of the breeding season on April 1st. Our model was structured with three age classes (adult, subadult, and juvenile) and two sexes. Juveniles transitioned to subadults at the start of their first non-breeding season in October, while subadults transitioned to adults at the start of their first breeding season in April. We modeled the population as closed to immigration and emigration as in previous bobwhite population models (Folk, Holmes & Grand, 2007; Sandercock et al., 2008; Rosenblatt et al., 2021), so that any change in abundance between years was a function of changes in fecundity and/or survival.

Figure 1 Directed acyclic graph (DAG) showing the relationship between data sources and population parameters of an integrated population model for northern bobwhites (Colinus virginianus) in southern Georgia, USA.

Data sources are denoted with boxes while parameters are denoted with circles. Arrows show dependencies between nodes in the model, while dashed arrows represent density-dependent effects. Submodels are denoted with rectangles. Subscript s denotes sex (male or female), m denotes breeding month (chicks hatched June–September), a denotes age (adult or subadult), and t denotes year. Notation: Na,s,t(Apr), abundance at the start of the breeding season in April; Nm,s,t(B) abundance at the start of breeding months with chick production (June–September); Cm,s,t number of chicks produced; N.Sm,s,t(Oct), number of young birds from each monthly cohort surviving to the start of the non-breeding season in October; N.As,t(Oct), number of adults surviving until the start of the non-breeding season in October, N.Sm,s,t(Nov), number of subadults from each monthly cohort surviving to November; N.As,t(Nov), number of adults surviving until November N.Sm,s,t(Dec), number of subadults from each monthly cohort surviving to December; N.As,t(Dec), number of adults surviving until December; Pm,s,t, per-capita productivity rates for each month June–September; Tm,s,t, number of adults from which productivity data was collected (supplied as data); ϕm,t(J), survival of juveniles from hatch month to the start of October; τt(Nov), ratios of ages, sexes, and subadults from each monthly cohort in November; τt(Dec), ratios of subadults from the June/July, August, and September monthly cohorts in December; ϕt(B), monthly survival during the breeding season (April–September); ϕa,s,t(NB), monthly survival during the non-breeding season (October–March); N.Tt, total number of individuals in November; Et; survey effort and average covey size relating abundance to covey counts (supplied as data); pt, detection process for covey counts (supplied as informative priors); CCt, November covey counts; Sa,s,t, survival data from radiotelemetry; BPm,s,t, breeding productivity from nestmonitoring data; CHt, ratios of ages, sexes, and subadults from each monthly cohort in the November trapping and harvest data; Ht, ratios of subadults from each monthly cohort in the December harvest data.

State process

We modeled yearly changes in population size as a function of changes in breeding and non-breeding survival of adults and subadults, juvenile survival, and per-capita productivity. As subadults transition to adults at the start of the breeding season, we only structured breeding parameters by sex, month, and year. We modeled abundance at the start of April ( N1,s,t(B), where one refers to the first month of the breeding season) for sex s in year t as the sum of the adults ( N1,s,t(Apr)) and subadults ( N2,s,t(Apr)) surviving from the previous winter. Breeding abundance at the start of subsequent months ( m) was a function of the abundance in the preceding month and the monthly survival rate during the breeding season ( ϕt(B)):

Nm,s,t(B)∼Binomial(Nm−1,s,t(B),ϕt(B)).

Too few males were tracked during the breeding season to estimate sex-specific breeding survival rates, though we note that many other bobwhite studies also assume no sex effects on survival during the breeding season (e.g., Sisson et al., 2009, Gates et al., 2012, Rectenwald et al., 2021).

We modeled chick production separately for each sex between June and September. The number of chicks produced by males and females in each of these months was modeled as a function of the monthly breeding abundance and per-capita productivity rate ( Pm,s,t):

Chicksm,s,t∼Poisson(Nm,s,t(B)∗Pm,s,t).

We assumed a 50/50 sex ratio of chicks, so that the number of male chicks was given by:

Chicksm,1,t(Sex)∼Binomial(sum(Chicksm,1:2,t),0.5)

and the number of female chicks ( Chicksm,2,t(Sex)) given by the difference between Chicksm,1,t(Sex) and sum(Chicksm,1:2,t). The number of juveniles surviving until recruiting as subadults at the start of the non-breeding season in October was a function of the number of chicks produced and the probability of juveniles from monthly cohort m surviving from breeding month m to the start of October ( ϕm,t(J)):

N.Subm,s,t(Oct)∼Binomial(Chicksm,s,t(Sex),ϕm,t(J)).

The overall survival rate ϕm,t(J) was calculated from juvenile daily survival rates ( ϕm,t(J.daily)) based on an average of 30.5-day months and the number of months between breeding month m and October. For example, the overall survival rate for the June cohort was given by ϕJun,t(J)=ϕJun,t(J.daily)∧(30.5∗4). In this way, we allowed juvenile survival to vary across monthly cohorts; however, we assumed that juvenile survival would be similar across sexes. Survival of juvenile bobwhite increases after the development of wing feathers 2-weeks post-hatch and again 1-month post-hatch when young become capable of thermogenesis (Lusk et al., 2005; Sandercock et al., 2008; Terhune, Chandler & Martin, 2017). We do not directly model these age-related changes in survival, and instead incorporate this variation into the single cohort-specific survival parameter. The ϕm,t(J.daily) parameter therefore represents the average daily survival rate of juveniles from the hatch month to the start of October in year t. Though we lack direct data to estimate this parameter, the differences between the ratio of chicks produced in each monthly cohort (dataset six) and the ratio of November and December survivors from each monthly cohort (datasets two, three, and four) should be reflective of differences in juvenile cohort survival.

We incorporated informative priors on ϕm,t(J.daily) to aid in model convergence. We modeled juvenile daily survival as arising from a Normal process with a global mean and standard deviation:

logit(ϕm,t(J.daily))∼Normal(logit(μm(phi.J.daily)),σ(phi.J.daily)).

We allowed the mean daily survival rate to vary by monthly cohort but assumed that the standard deviation would be similar across cohorts. Values of μm(phi.J.daily) and σ(phi.J.daily) were derived from the mean daily survival rates of monthly cohorts and the variation in overall daily survival rates across years reported by Terhune, Palmer & Wellendorf (2019). This study used recoveries of patagial-tagged chicks to estimate juvenile bobwhite survival at a study site that was <100 km from our study area (Terhune, Palmer & Wellendorf, 2019). We used values of 0.9866, 0.9911, 0.9902, and 0.9928 for daily survival of the June–September cohorts, respectively, and specified σ(J.daily) to take the value of 0.551. A prior sensitivity analysis indicated that posterior distributions of ϕm,t(J.daily) were sensitive to the prior specification (Figs. S1–S4) but, with the exception of the fully vague prior, generally had little effect on yearly abundance estimates (Fig. S5, Table S1).

To facilitate modeling of density-dependent effects during the non-breeding season on survival and capture (November) and harvest (November and December) data (datasets three and four), we estimated the age-and sex-structured population size in each month October–December ( k). Monthly non-breeding adult abundance was calculated as the number of survivors from the previous month:

N1,s,t(k)∼Binomial(Ns,t(k−1),ϕ1,s,t(NB))

where ϕ1,s,t(NB) is the monthly non-breeding survival rate of adults and the 1 subscript denotes adults. For the first month of the non-breeding season (October), Ns,t(k−1) refers to the sex-specific abundance at the start of September and ϕt(B) was used instead of ϕ1,s,t(NB). The number of surviving subadults from each monthly cohort in November and December was modeled as:

N.Subm,s,t(k)∼Binomial(N.Subm,s,t(k−1),ϕ2,s,t(NB))

where ϕ2,s,t(NB) is the monthly non-breeding survival rate of subadults. The number of subadults October–December ( N2,s,t(k)) was then calculated as the sum of N.SubJun−Sep,s,t(k) across monthly cohorts. Finally, the number of individuals surviving over the winter from December to the start of breeding in April was given by:

Na,s,t+1(Apr)∼Binomial(Na,s,t(Dec),ϕa,s,t(NB))

where the subscript a denotes the age class (adults or subadults).

We constrained yearly non-breeding survival to be normally distributed on the logit-scale around a global mean ( ϕ.μa,s(NB)), corresponding to a random year effect. We also incorporated a density-dependence parameter ( γ(NB)) denoting the effect of October population size ( sum(N1:2,1:2,t(Oct))) on ϕa,s,t(NB) via the following equations:

ϕ.logita,s,t(NB)∼Normal(logit(ϕ.μa,s(NB)),ϕ.σa,s(NB))

logit(ϕa,s,t(NB))=ϕ.logita,s,t(NB)+γ(NB)∗sum(N1:2,1:2,t(Oct)).

We used a similar process to model yearly variation in breeding survival and monthly rates of per-capita productivity with slight modifications. Breeding survival was modeled with density-dependent effects of April population size ( ϕ.logitt(B)+γ(B)∗sum(N1,1:2,t(B))), while productivity was modeled with density-dependent effects of monthly population size P.logm,s,t+γ(prod)∗sum(Nm,1:2,t(B))) and was modeled on the log scale rather than the logit scale. We constrained density-dependent effects to be non-positive as we had no a priori reason to expect allee effects in the population. In a preliminary analysis, we found that the model sometimes had difficulty estimating male productivity rates due to the low number of tracked males in some years. Males generally brood fewer nests than females (Sandercock et al., 2008); therefore, we constrained male productivity to be less than or equal to female productivity in each month and year.

Observation processes

We modeled the trapping and harvest data as realizations of Multinomial distributions with vectors of probabilities derived from the state process. First, we modeled the number of adult males, adult females, and subadults captured in November (dataset one) via a multinomial distribution with probabilities of N1,1,t(Nov)/sum(N1:2,1:2,t(Nov)), N1,2,t(Nov)/sum(N1:2,1:2,t(Nov)), and sum(N2,1:2,t(Nov))/sum(N1:2,1:2,t(Nov)), respectively. Similarly, the number of subadults from monthly cohorts (June–September) in the November trapping (dataset two) and harvest (dataset three) were specified with probabilities equal to (NJun−Sep,1,t(Nov.Sub)+NJun−Sep,2,t(Nov.Sub))/ sum(N2,1:2,t(Nov)). The December harvest data (dataset four) only contained three categories (June/July cohort, August cohort, September cohort) because subadults from the June and July cohorts cannot be reliably separated past late November. For this last distribution, we specified probabilities similar to November except for the probability of the June/July cohort specified as the ratio of the male and female subadults from the June and July cohorts to the total subadults in December.

Age-and-sex ratios may not reflect population ratios if certain age-or-sex classes are disproportionately vulnerable to capture or harvest (Zimmerman et al., 2010; Arnold, 2018); though we believe this to be unlikely in our study system. Bobwhites form coveys of mixed sexes and ages in the fall and so are unlikely to show strong capture bias between ages or sexes. Marsden & Baskett (1958) reported no age bias in bobwhite trapping, so we took the approach of other studies modeling bobwhite abundance (Nolan et al., 2023) and assumed no bias in capture probability across ages and sexes. Age-and-sex classes are known to be disproportionately vulnerable to harvest (Pollock et al., 1989); however, we only used harvest data to assess the ratio of juveniles of different monthly cohorts. Pollock et al. (1989) found no difference in harvest vulnerability between juvenile males and females; however, even if these classes do differ in harvest vulnerability in our study system, the model was structured with a 50/50 sex ratio for each monthly cohort. Any bias in harvest vulnerability should therefore be shared across monthly cohorts.

We modeled bi-weekly survival of bobwhites throughout the annual cycle (dataset five) using a known-fate model (White & Garrott, 1990). We calculated bi-weekly survival estimates for the observation process as the square root of the monthly survival estimates from the state process. We then modeled the binary alive/dead state of individual i in bi-weekly period q ( zi,q) over the course of the tracking period (e.g., first bi-weekly period after transmitter deployment until either confirmed mortality or censoring) as:

zi,q∼Bernoulli(zi,q−1∗ϕa−1,s,b−1,t−1(biweek))

where ϕa−1,s,b−1,t−1(biweek) represents the bi-weekly survival rate from period q−1 to q indexed by age ( a), sex ( s), season ( b), and year ( t) in q−1. Similar to previous analyses of bobwhite telemetry data (e.g., Howell, Terhune & Martin, 2021, Rosenblatt et al., 2021), this method assumes that detection of birds with active transmitters will be near 100%.

The total number of chicks produced from adult males and females in each month (dataset six) was modeled as a realization of a Poisson process based on the per-capita productivity rate ( Pm,s,t, latent variable) and the number of adults of sex s tracked in month m ( Trackedm,s,t, observed data). The model requires accurate estimates of chick production; however, 17% of successful nests did not have information recorded about the number of chicks hatched. We performed a preliminary analysis to estimate the number of chicks hatched from nests with missing data (Article S1) and used the mean ( μm,s,t(Chicks.est)) and standard deviation ( σm,s,t(Chicks.est)) of posterior samples for the number of chicks produced for a given sex, month, and year in the IPM. We modeled μm,s,t(Chicks.est) as arising from a Normal distribution around the true number of chicks produced ( Tot.Chicksm,s,t):

μm,s,t(Chicks.est)∼Normal(Tot.Chicksm,s,t,σm,s,t(Chicks.est)).

We then modeled Tot.Chicksm,s,t as arising from a Poisson distribution based on the per-capita productivity rate and the number of tracked adults:

Tot.Chicksm,s,t∼Poisson(Trackedm,s,t∗Pm,s,t).

The number of coveys detected on fall grid surveys (dataset seven) is a function of the true abundance across surveyed grids, the average covey size, and the detection process. As only a proportion of grids were surveyed in each year ( Effortt), we modeled the total number of bobwhites on sampled grids as:

Nt(Grid)∼Binomial(sum(N1:2,1:2,t(Nov)),Effortt).

Covey size was estimated at a subset of grids each year. We modeled uncertainty in average covey size by:

CSt(Act)∼Normal(μt(CS),σt(CS))

where CSt(Act) is the true average covey size in year t, μt(CS) is the observed mean covey size across grid cells in year t, and σt(CS) is the observed standard deviation across grids in year t. We bounded estimates of CSt(Act) to be within minimum and maximum values, which we specified based on the 2.5% and 97.5% quantiles of the observed covey counts in grid cells (10 and 18, respectively). We then converted between abundance and number of coveys on sampled grids by:

Nt(Coveys)∼Poisson(Nt(Grid)/CSt(Act)).

The detection process can be decomposed into two processes (Amundson, Royle & Handel, 2014). The first is availability ( Availabilityt(Act)), which is the probability of a bobwhite covey calling during the survey and thus being able to be detected. Second, conditional detection ( p.det(Act)) is the probability of observers hearing and recording a covey, conditional on it being available. We modeled the number of coveys detected on grid surveys with a Binomial distribution:

Covey.Countt∼Binomial(Nt(Coveys),Availabilityt(Act)∗p.det(Act)).

No data were available to estimate Availabilityt(Act) or p.det(Act), so we incorporated these parameters into the model using informative priors based on previously-published bobwhite literature (Article S2).

Model fitting and follow-up analyses

We implemented the IPM within a Bayesian framework using the NIMBLE package (de Valpine et al., 2017) in R v 4.1.1 (R Core Team, 2021). Posterior samples were drawn from three chains of Markov Chain Monte-Carlo (MCMC) simulations implemented with 120,000 iterations and a burn-in phase of 30,000 iterations. We incorporated vague uniform or gamma priors on parameters ( ϕ.logitt(B), P.logm,s,t, ϕ.logita,s,t(NB), and associated global parameters, N1:2,1:2,1(Apr), γ(NB), γ(B), γ(prod)), except for those specifically mentioned above. Model convergence was assessed through visual inspection of MCMC chains. We provide parameter estimates in Table S1.

As a follow-up analysis, we used several forms of retrospective analysis to assess the relative importance of demographic drivers to changes in annual population growth rate ( λt=Nt+1(Apr.Tot)−Nt(Apr.Tot)). First, we assessed the average of vital rate posterior samples from the MCMC across years with positive population growth rates and across years with negative population growth rates. We identified years of positive and negative growth rates as those where the 95% credible intervals of λt were >1 and <1, respectively. Second, we used the posterior samples to assess the correlation between each demographic rate and population growth (McConnell et al., 2018; Saunders et al., 2019). As a more formal analysis, we used transient life-table response experiments (tLTREs) to decompose changes in the realized population growth rate due to changes in demographic rates and population structure over time (Koons et al., 2016; Koons, Arnold & Schaub, 2017). In contrast to traditional life-table response experiments (LTREs) (Caswell, 1989, 2010), tLTREs model transient dynamics (i.e., the population is not assumed to be at a stable state) by explicitly incorporating variation in the population stage structure over time. We used two variants of the tLTRE described in Koons, Arnold & Schaub (2017) and Schaub & Kéry (2021), which correspond to the random-design and fixed-design LTREs of Caswell (2001). The random-design approach decomposes the variance of the realized growth rate due to the variance and covariance of the demographic rates and population age and sex structure while the fixed-design approach assesses how the change in realized growth rate between successive years were driven by changes in demographic rates and population structure. Both tLTRE variants were implemented based on code in Schaub & Kéry (2021) and described in Article S3.

Results

Data summary

We used radiotelemetry to track 4,844 bobwhites (3,397 females and 1,447 males) over the 25-years of the study. A small proportion of birds (0.6%) were tracked until radio failure, were recaptured at a later period, and were re-deployed with a second radiotransmitter. The majority (61%) of deployments terminated in mortality; of these, 59% was attributable to avian predation (hawks and owls), 13% to mammalian predation, 4% to harvest, 2% to snake predation, and 22% to a different or unknown source of mortality. We found and monitored 22 to 115 bobwhite nests each year, totaling 1,608 nests over the entire course of the study. Only 5% of nests were attended by a male. The majority of nests (52%) fledged at least one chick, though the success rate varied yearly from 30% to 69%. Successful nests fledged 10.3 chicks on average (sd: 3.6). We detected between 5.5 and 13.3 bobwhite coveys per survey grid each year. We captured and banded 4,317 bobwhites across the November trapping seasons, with only 1.3% of banded birds being recaptured in a subsequent season. Fall captures were composed of 10% adult females, 17% adult males, and 73% juveniles, with 28%, 32%, 29%, and 12% of juveniles being aged to the June–September cohorts, respectively. Across the entire study period, 4,619 juveniles were harvested in the early winter (1,841 in November and 2,778 in December). The proportion of juveniles from the June–September cohorts in the November harvest data was 49%, 21%, 18%, and 12%, respectively. In December, 75% of harvested juveniles were from the June-July cohort, 19% were from the August cohort, and 6% were from the September cohort.

Population demographics

In the following section, model estimates are reported as median and 95% credible intervals, unless otherwise stated. Bobwhite population density fluctuated but was relatively stable from 1998–2012, though the population generally increased over the last decade of the study (Fig. 2). Over the course of the study, spring population density increased from 3.02 birds/ha in 1998 (2.03–4.38) to 4.42 birds/ha in 2022 (3.78–5.16). This corresponds to a change in total spring abundance from 906 birds in 1998 (608–1,314) to 1,327 birds in 2022 (1,134–1,547).

Figure 2 Model estimates of population density (birds/ha) of Northern Bobwhites (Colinus virginianus) in southern Georgia, USA, 1998–2022.

Density is shown for the population in April (black triangles) and November (orange circles). Mean estimates and 95% credible intervals are shown.

The global mean of adult breeding survival from April–September was 0.48 (0.41–0.61), though realized values ranged from 0.26 (0.20–0.33) to 0.57 (0.50–0.64) (Fig. 3A). As with temporal trends in density, adult breeding survival increased after 2014. Non-breeding survival from October–March was generally higher than during the breeding season, with global means of 0.66 (0.61–0.73), 0.66 (0.58–0.75), 0.62 (0.57–0.70), and 0.68 (0.62–0.76) for adult male, adult female, subadult male, and subadult female 6-month non-breeding survival, respectively (Fig. 3B). The one exception was in 1999, when all age-and-sex-classes except adult males exhibited reduced survival. The non-breeding survival global means were largely similar between ages and sexes; however, males, and especially adult males, generally experienced less annual variation in realized non-breeding survival rates than did females.

Figure 3 Model estimates of overall survival during the breeding (Apr–Sep, A) and non-breeding season (Oct–Mar) for Northern Bobwhites (Colinus virginianus) in southern Georgia, USA, 1998–2022.

The model estimated monthly survival rates, which were raised to the sixth power to calculate survival over the breeding/non-breeding season. Survival was estimated separately for adult males (black circles), adult females (green downward triangles), subadult males (upward orange triangles), and subadult females (blue squares) during the non-breeding season. Survival was assumed to be similar between males and females during the breeding season (subadults transition to adults at the start of the breeding season). Mean estimates and 95% credible intervals are shown.

Monthly per-capita productivity rates were higher for females than for males and declined throughout the breeding season (Fig. 4A). Realized estimates of total per-capita productivity (total number of chicks produced per bird summed across June-September) fluctuated annually (Fig. 4B), with female productivity ranging from 2.51 chicks/female in 2014 (2.07–3.00) to 6.37 chicks/female in 2001 (5.51–7.28). Female-attended nests were responsible for an average of 82% of chicks produced during the study (80–84%), though this percentage varied annually from 51% (38–65%) to 98% (94–99%). Per-capita productivity generally decreased over the first decade of the study period, before increasing in 2007, and declining again from 2013–2014. This decline in productivity in 2013–2014 was only observed in females and not in males.

Figure 4 Model estimates of sex-specific per-capita productivity for Northern Bobwhites (Colinus virginianus) in southern Georgia, USA, 1998–2022.

Estimates from males are denoted with black circles while estimates from females are denoted with orange downward triangles. Estimates show the global means for the sex-specific monthly per-capita productivity June–September (A) and the annual number of chicks produced during an entire breeding season divided by the number of breeding adults of each sex (B). For comparison with other studies, we also show the total number of chicks produced in a breeding season divided by the number of breeding females (blue diamonds) in figure B. Mean estimates and 95% credible intervals are shown.

Juvenile daily survival was similar across monthly cohorts, with mean estimates of 0.991 (0.988–0.995) from the June cohort, 0.992 (0.988–0.995) from the July cohort, 0.990 (0.983–0.994) from the August cohort, and 0.993 (0.989–0.995) from the September cohort. As earlier cohorts were required to survive for a greater duration to reach the end of the breeding season, overall cohort survival to the start of October decreased from June to September (Fig. 5). The degree to which juvenile survival estimates were affected by the informative priors varied by monthly cohort. Model estimates of June cohort survival were consistently higher than the 95% credible intervals of the prior distribution (Fig. 5A) and increased after 2014 much like adult survival. Later cohorts became more-and-more similar to the priors, with estimates from the September cohort nearly completely conforming to the prior distribution (Figs. 5B–5D, Figs. S6–S9).

Figure 5 Model estimates of juvenile cohort survival for a population of Northern Bobwhites (Colinus virginianus) in southern Georgia, USA, 1998–2022.

Cohort survival is defined as survival from the hatch month until the start of the non-breeding season in October. Juvenile survival estimates are shown for cohorts of birds hatched in June (A), July (B), August (C), and September (D). Mean estimates and 95% credible intervals are shown. Horizontal dashed lines represent the upper and lower 95% quantiles for the informative prior distributions placed on juvenile cohort survival parameters. Informative priors were based on cohort-specific survival rates reported by Terhune, Palmer & Wellendorf (2019).

Population density had only minor effects on breeding survival and non-breeding survival but had more substantial negative effects on per-capita productivity. For every one additional bird in the breeding population, monthly female per-capita productivity was estimated to decline by 0.0008 (0.0004–0.0012) chicks/female. This corresponds to a relative decline in productivity of 0.07% across the breeding season (0.04–0.1%). In contrast, an increase of one bird in the population only resulted in relative declines of 0.008% (0.0003–0.04%) and 0.002% (0.00007–0.008%) in breeding and non-breeding 6-month survival, respectively. The realized median April abundance estimates over the course of the study ranged from 642 to 1,384, corresponding to a change in per-capita female productivity from 5.85 chicks/female (5.34–6.40) to 3.39 chicks/female (2.31–4.40) due to negative density-dependent effects. As a post-hoc analysis, we performed a correlation analysis between posterior samples of annual April population density and three components of annual productivity calculated from the observed data: the proportion of radiotracked females surviving the breeding season which initiated nests, the proportion of breeding females surviving the breeding season which initiated renest attempts following failure of first broods, and the proportion of breeding females surviving the breeding season which initiated double-brood attempts following successful first broods. We only focused on females for this post-hoc analysis because very few males were tracked with radiotelemetry in some breeding seasons and almost all males attempted only one brood/year. All three components of annual productivity were negatively correlated with April density (Fig. S10; nest initiation probability r: −0.28, −0.46 to −0.04; renest probability r: −0.34, −0.50 to −0.19; double-brood probability r: −0.39, −0.53 to −0.24).

Demographic drivers of population growth

The correlation analysis revealed that female productivity in June was most correlated with variation in spring population growth (r: 0.48, 0.29–0.65), followed by female productivity in September (r: 0.41, 0.24–0.56), male productivity in August (r: 0.30, 0.07–0.53), and female productivity in August (r: 0.26, 0.02–0.49) (Fig. S11). Subadult female non-breeding survival (r: 0.45, 0.22–0.64), adult breeding survival (r: 0.33, 0.11–0.52), and adult female non-breeding survival (r: 0.32, 0.07–0.52) were also correlated with spring population growth, though these results were largely driven by 1–2 years with very low survival and low population growth. Comparing average vital rates between increasing and decreasing years (i.e., λt > 1 and λt < 1, respectively) yielded similar results to the correlational analysis (Table S2). Results from the tLTRE analyses were also generally similar to the correlational analysis. For the random-design approach, breeding survival had the highest total contribution to variation in population growth over the entire study period, followed by subadult female non-breeding survival, June female productivity, and September female productivity (Fig. 6). Juvenile survival and proportional population structure had little effect on bobwhite population growth. Decomposing the change in population growth rate between successive years into contributions from changes in demographic rates (fixed-design approach) revealed that the dominant drivers of population dynamics varied across the study duration (Fig. 7). Non-breeding survival generally had little effect on changes in population growth except in the first few years of the study, corresponding to the large decrease in non-breeding survival for most age-and-sex classes in 1999. Breeding survival also had little effect on changes in population growth except around the years of population transition around 2013 and 2014. In contrast, changes in per-capita productivity generally had large contributions to the change in population growth across the entire study duration.

Figure 6 Results of a random-design transient life-table response experiment (tLTRE) for a population of Northern Bobwhites (Colinus virginianus) in southern Georgia, USA, 1998–2022.

The random-design tLTRE assesses the total contribution of realized variation in demographic rates and proportional population structure to variation in the spring population growth rate. Parameters incorporated into the model were: daily survival of juveniles from the June (Jun J Surv), July (Jul J Surv), August (Aug J Surv), and September (Sep J Surv) cohorts, adult breeding survival (B Surv), non-breeding survival of adult males (A M NB Surv), adult females (A F NB Surv), subadult males (S M NB Surv), and subadult females (S F NB Surv), per-capita productivity rates for males and females in June (Jun M Prod/Jun F Prod), July (Jul M Prod/Jul F Prod), August (Aug M Prod/Aug F Prod), and September (Sep M Prod/Sep F Prod), and the ratio of adult males (N A M), adult females (N A F), subadult males (N S M) and subadult females (N S F) surviving to the start of the breeding season in April. Bars show mean estimates while vertical lines show the upper and lower 95% credible intervals of estimates.

Figure 7 Results of a fixed-design transient life-table response experiment (tLTRE) for a population of Northern Bobwhites (Colinus virginianus) in southern Georgia, USA, 1998–2022.

The fixed-design tLTRE assesses contributions of the changes in demographic rates and proportional population structure between successive years on changes in spring population growth rates. The analysis was performed using all 21 parameters from Fig. 6, though for ease of interpretation we have aggregated the contribution absolute mean values into five categories: population structure (summed contributions of ratio of birds in each age-and-sex class surviving to the start of breeding in April, black bars), adult breeding survival (light gray bars), non-breeding survival (summed contributions of age-and-sex-specific non-breeding survival rates, white bars), juvenile survival (summed contributions of June–September cohort daily survival rates, dark gray bars), and productivity (summed contributions of sex-specific per-capita productivity rates for each month June–September, dashed bars).

Discussion

This study is part of a growing body of research showing that IPMs can be used to effectively monitor population dynamics of Northern Bobwhites (McConnell et al., 2018; Rosenblatt et al., 2021; Sinnott et al., 2023). Our population of bobwhites in southern Georgia has been stable or increasing in abundance over the past 25 years. Though bobwhite populations have been broadly declining across the range, including in the southeastern US (Hernández et al., 2013; Sauer, Link & Hines, 2020), our results are consistent with previous findings from the region that populations can be stable in areas of intensive bobwhite management (McConnell et al., 2018).

The study population showed signs of population regulation before 2013, as spring density oscillated around three birds/ha; however, population dynamics underwent a notable shift starting in 2013–2014. After these years, breeding survival increased, per-capita female fecundity decreased, and population density started to increase. This change in population dynamics may be explained by management activities and changes in the predator community during this period. Trapping removal of mammalian predators, bobcats (Lynx rufus) and coyotes (Canis latrans), increased during this time period. Furthermore, abundance of two major bobwhite predators, Cooper’s (Accipiter cooperii) and sharp-shinned hawks (Accipiter striatus), declined at the study site after 2014 (Rectenwald et al., 2021; Bellier et al., 2023). This hawk decline was likely associated with broad-scale closed canopy early-rotation pine thinning in the surrounding area (C Sisson, 2024, personal observations) reducing the habitat quality for Accipiters (Marzluff et al., 2002). Bobwhites may be especially vulnerable to hawk predation during the breeding season (Rectenwald et al., 2021), potentially explaining why breeding survival increased post-2014 but non-breeding survival was unaffected. Increases in breeding bobwhite density may then have dampened productivity through density-dependent processes. Continued monitoring is needed to determine if this change in population dynamics represents a new carrying capacity or if dynamics will eventually shift back to pre-2013 levels.

Previous research has found that the demographic drivers of population growth in bobwhites varies regionally. Studies from southwest Missouri and the southeastern US have found population growth to be most affected by changes in productivity (McConnell et al., 2018; Sinnott et al., 2023), while studies from Ohio, Wisconsin, and from demographic data collected from across the range have found growth to be most affected by non-breeding survival (Folk, Holmes & Grand, 2007; Sandercock et al., 2008; Gates et al., 2012; Rosenblatt et al., 2021). This regional variation may be partially attributable to differences in model design, particularly in terms of how breeding productivity is modeled; however, the demographic drivers of local bobwhite population growth may also be affected by population size and trajectory. Studies identifying non-breeding survival as the dominant driver of growth rates generally come from low-density and declining populations (Gates et al., 2012; Rosenblatt et al., 2021; Sandercock et al., 2008; Williams et al., 2012, but see Sinnott et al., 2023), while empirical (McConnell et al., 2018) and simulation (DeMaso et al., 2011) studies have identified productivity as the dominant driver in higher density populations that are stable or increasing. Variation in the drivers of population growth may also be affected by regional differences in limiting factors. For example, non-breeding survival is generally the dominant driver of population growth rates at the northern edge of the range (Folk, Holmes & Grand, 2007; Gates et al., 2012; Williams et al., 2012; Rosenblatt et al., 2021), where populations are often limited by the effects of winter temperature and snowfall on non-breeding survival (Janke et al., 2017; Wolske, Behney & Powell, 2023). In contrast, productivity has been found to be the dominant driver of population growth rates further south in the range where winter conditions are less severe (McConnell et al., 2018; Sinnott et al., 2023, though see results from Alabama in Folk, Holmes & Grand, 2007). Our study population exhibited a combination of findings from other parts of the bobwhite range; per-capita productivity was a major contributor to variation in population growth rates over the entire study period, while breeding and non-breeding survival had large effects on population growth rates in a few years of the study. Assessing the demographic contributions to yearly changes in population growth (fixed-effect tLTRE) allowed us to clarify this pattern, which would have been missed if only using long-term averages to assess the demographic drivers of population growth (random-effect tLTRE). Wildlife managers generally aim to maintain or promote population abundance (i.e., desire λ ≥ 1); necessitating accurate identification of the vital rates driving population growth rates. The demographic drivers of population dynamics can shift over time in relation to changing environmental conditions (Caswell, 2007; Maldonado-Chaparro et al., 2018); therefore, we suggest that researchers should carefully consider annual variation when performing retrospective population analyses to avoid basing management decisions off erroneously-identified drivers of population growth rates.

Besides estimating the historical demographic drivers of population growth, we found that our study population was strongly affected by density-dependent effects on female productivity, with limited effects on survival or male productivity. This result is consistent with previous work in the southeastern US showing negative density-dependent effects on reproduction and recruitment (McConnell et al., 2018; Palmer, Wellendorf & Sisson, 2022). The productivity term in our IPM encompassed multiple reproductive parameters. though our post-hoc analysis (Fig. S10) suggested that the negative density-dependent effects on productivity were, at least partially, driven by changes in the rates at which female bobwhites initiated breeding, renesting, and double-brooding. Density-dependent processes appear to act most strongly on female productivity in the southeastern US, though density-dependent processes may vary regionally in responses to differences in environmental conditions and limiting factors (Stiling, 1988; Wang et al., 2006; Hixon et al., 2012). While integrated population models have been suggested as a powerful tool for directly modeling density-dependent effects on vital rates (Besbeas et al., 2003; Abadi et al., 2012), relatively few studies have incorporated density-dependence, much less multiple density-dependent pathways, in such a way (Schaub & Kéry, 2021). Our modeling approach assumed that all density-dependent effects are equal across all individuals in the population (i.e., a single density-dependence parameter for ϕa,s,t(NB), ϕt(B), and Pm,s,t); however, certain age and sex classes often can respond or contribute differently to density-dependent effects on vital rates (Bonenfant et al., 2009; Gamelon et al., 2016; McFarlane et al., 2022). Future work could expand upon our modeling framework by allowing density-dependent effects on non-breeding survival to vary based on age and sex classes. Furthermore, future research could examine the relative contribution of each age/sex class to density-dependent effects as in Gamelon et al. (2016). Assessing the relative strengths and mechanisms underlying density-dependent processes of target populations is a necessary component for setting harvest policies and understanding population regulation (Hixon, Pacala & Sandin, 2002; Guthery & Shaw, 2013; Péron, 2013). Previous studies have assessed density-dependence in bobwhite populations through annual population fluctuations (Williams, Ives & Applegate, 2003; Guthery & Shaw, 2013), simulations (DeMaso et al., 2011, 2013), or modeling density-dependent effects on a single vital rate (McConnell et al., 2018); however, our study is the first to jointly estimate density-dependent effects on multiple vital rates in a single population model. Our results show that IPMs provide ideal frameworks for simultaneously and directly modeling density-dependent processes on all relevant population vital rates.

Our modeling framework also utilized a novel method of combining breeding and post-breeding data in an IPM framework to infer juvenile survival in the absence of direct data. This method was moderately informative for inferring juvenile survival but required informative priors to aid in model convergence. Previous studies have found that parameters inferred by IPMs without direct data are often estimated with low precision (Plard et al., 2019; Rosenblatt et al., 2021), thus necessitating an informative prior. Alternatively, cohort survival rates were the product of 30.5–122 daily survival rates (depending on hatch month), such that a vague Normal prior on the logit scale could substantially bias abundance at the cohort survival scale (Northrup & Gerber, 2018). Still, integrating monthly productivity, post-breeding count, and post-breeding capture and harvest ratio data successfully updated the prior distributions. This effect was mainly observed with the June cohort of chicks, likely because this cohort had the longest juvenile stage. Furthermore, more chicks were hatched in June (47%) than in other months (8–26%). June juvenile survival estimates increased throughout the study period and were generally higher ( μJun(phi.J.daily): 0.992, 0.988–0.995) than reported by Terhune, Palmer & Wellendorf (2019) from northern Florida (0.9866). The higher June survival observed in our study could reflect differences in site quality, such as an emphasis on brood habitat management, supplemental feeding occurring throughout the year, or reduced predation risk from declining hawk populations at our study site (Bellier et al., 2023). Our informative prior was based on a capture-recapture study of juveniles marked with patagial tags, potentially suggesting that effects of capture or tag attachment may negatively affect juvenile survival (Trefry, Diamond & Jesson, 2013). Juvenile survival had little effect on historical population growth rates compared to the other vital rates, perhaps because this parameter was only indirectly inferred while the others were directly modeled with data. Additionally, juvenile survival exhibited relatively little variation across time and so would have little effect on variation in realized population growth rates. Juveniles of most species cannot be aged to specific monthly cohorts as can bobwhites, though the same methods could be applied by integrating seasonal fecundity, post-breeding counts, and post-breeding age-ratios data to help estimate juvenile survival. Estimation of latent vital rates without explicit data in an IPM framework is dependent on model and prior specification and can be heavily biased if model assumptions are violated (Riecke et al., 2019; Paquet et al., 2021). Still, many population models lack direct data on specific vital rates, such as juvenile survival, even though accurately and precisely estimating these parameters can be important for mechanistic population models and for assessing the demographic drivers of population growth (Fefferman & Reed, 2006; Koons et al., 2016). Our results suggest that combining productivity and post-breeding data with prior information could be useful for improving estimates of juvenile survival in an IPM framework, though researchers must always be cautious when interpreting latent variables estimated without explicit data.

Despite sampling the same population parameter, age ratios among monthly juvenile cohorts differed between the capture and harvest datasets, i.e., the ratio of June birds was much higher in the November harvest data (49%) than the capture data (28%). Using an IPM framework allowed us to directly incorporate these conflicting datasets rather than omitting or statistically downweighing certain datasets (Maunder & Piner, 2017; Saunders et al., 2019). While we had no a priori reason to expect bias in the monthly cohort ratios, the conflicting data suggest that bias may exist in at least one dataset. The methods used to age birds likely introduced some uncertainty (e.g., error in assessing molt or in the backdating calculation to estimate hatch month); however, measurements were performed by experienced observers and any error would likely be randomly distributed across birds rather than strongly biased towards a particular dataset. Instead, birds from different monthly cohorts may vary in their capture and/or harvest probability.

These biases can be assessed either through the analysis of within-season recaptures (capture bias, Arnold, 2018) or assessing ratios of banded birds harvested in the subsequent hunting season (harvest bias, Zimmerman et al., 2010). We performed a post-hoc analysis using a binomial generalized linear model to assess capture bias, utilizing yearly values for the number of released and harvested banded subadult birds from each monthly cohort. Our results should be interpreted with caution, as only an average of 3.8 banded subadults/year were harvested, but the 95% credible intervals of harvest rate estimates were overlapping for all monthly cohorts (June: 0.017–0.036, July: 0.027–0.047, August: 0.025–0.046, September: 0.017–0.050). Age ratios may instead be biased in capture data. Specifically, capture rates for the June cohort could be biased low if older subadults (i.e., from earlier monthly cohorts) were more trap-wary or were in better physical condition and so less likely to enter baited traps (Weatherhead & Greenwood, 1981; Senar et al., 1999). We do not possess the data for a formal statistical test of capture bias among monthly cohorts, as researchers generally only recorded the first capture of each bird during the trapping season. We suggest that researchers should collect within-season recapture data to assess capture bias, and to directly account for capture and harvest bias between ages and sexes into future modeling efforts (Zimmerman et al., 2010; Amrhein et al., 2012; Arnold, 2018; Roberts et al., 2024).

Conclusions

Our study shows that IPMs can be a valuable tool for estimating vital rates, density-dependent processes, and the demographic drivers of population growth for bobwhites. Our bobwhite study population exhibited stable-to-increasing density over time, with variation in population growth rates driven largely by variation in productivity and, in a few years, survival. Though requiring long-term demographic data to develop the IPM, many of our findings would have been difficult or impossible to obtain if analyzing each dataset independently rather than in an integrated framework. Our IPM framework allowed for direct and simultaneous assessment of the relative strength of density-dependence on multiple demographic pathways and for combining breeding and post-breeding data to infer rates of juvenile survival in the absence of direct data. Future work could expand on our framework by allowing for density-dependent effects to vary by age and sex or by incorporating emigration and immigration into the population model. We suggest that implementing these methods could be broadly useful for improving population modeling not just for bobwhites, but also for a wide variety of study systems where adequate data exists.

Supplemental Information

Supplemental Information 1 Parameter estimates of population dynamics for a population of northern bobwhites (Colinus virginianus) in southern Georgia, USA, 1998– 2022.

The lower (2.5%), median (50%), and upper (97.5%) quantiles of posterior samples are shown for each year of the study. D refers to density (birds/ha), and λ refers to population growth rate from year to year+1 for the bobwhite population at the start of April (Apr.Tot superscript) and November (Nov.Tot superscript). ϕ^((J.daily)) is the daily juvenile survival rate, 〖ϕ.μ〗^((B))and ϕ^((B)) are the global mean and yearly (respectively) bi-weekly survival rate during the breeding season (April–September), 〖ϕ.μ〗^((NB))and ϕ^((NB)) are the global mean and yearly (respectively) bi-weekly survival rate during the non-breeding season (October–March), P.μ and P are the global mean and yearly (respectively) per-capita productivity rate, R is the ratio of age and sex classes in November (adult males, adult females, subadults), 〖CS〗^((Act)) is the average covey size, 〖Availability〗^((Act)) is the probability that bobwhite coveys would actively call and be available for detection during November covey count surveys, 〖p.det〗^((Act)) is the probability (conditional on availability) of at least one observer detecting a calling covey during November covey count surveys, 〖γ〗^((B)) is the effect of April density on adult breeding survival, γ^((NB)) is the effect of October density on non-breeding survival, and γ^((prod)) is the effect of monthly density (June–September) on monthly per-capita productivity rates. Subscripts for vital rates correspond to month of the breeding season (m, 1 = June, 2 = July, 3 = August, 4 = September), age during the non-breeding season (a, 1 = adult, 2 = subadult), and sex (s, 1 = male, 2 = female). Estimates are shown for the main model reported in the text using an informative prior on ϕ^((J.daily)) (Informative Prior) and for a model using a vague prior on ϕ^((J.daily)) (Vague Prior).

Supplemental Information 2 Summary of average vital rates of northern bobwhites (Colinus virginianus) calculated across posterior samples from years with positive and negative population growth rates.

Posterior samples were generated from an Integrated Population Model using demographic data from a bobwhite population in southern Georgia, USA, 1998–2022. Years were classified as positive (2001, 2012, 2015, 2019) or negative (1999, 2004, 2006, 2014) if the 95% credible intervals of the estimated April population growth rate ( <!--[if !msEquation]--> <!--[endif]-->) were greater or lesser than 1, respectively. Estimates are show for: April population density (birds/ha, <!--[if !msEquation]--> <!--[endif]-->), bi-weekly breeding survival ( <!--[if !msEquation]--> <!--[endif]-->), bi-weekly non-breeding survival ( <!--[if !msEquation]--> <!--[endif]-->), daily juvenile survival ( <!--[if !msEquation]--> <!--[endif]-->), monthly per-capita productivity ( <!--[if !msEquation]--> <!--[endif]-->), and total per-capita productivity ( <!--[if !msEquation]--> <!--[endif]-->). Subscripts for vital rates correspond to month of the breeding season (m, 1 = June, 2 = July, 3 = August, 4 = September), age during the non-breeding season (a, 1 = adult, 2 = subadult), and sex (s, 1 = male, 2 = female). Parameters that differ between years of increasing and decreasing population growth are bolded.

Supplemental Information 3 Methods of accounting for missing productivity data for estimating demographic parameters of northern bobwhites (Colinus virginianus) in southern Georgia, USA, 1998–2022.

Supplemental Information 4 Methods of implementing informative priors on the abundance detection process for estimating demographic parameters of northern bobwhites (Colinus virginianus) in southern Georgia, USA, 1998–2022.

Supplemental Information 5 Methods of performing life-table response experiments based on posterior samples from integrated population models of northern bobwhites (Colinus virginianus) in southern Georgia, USA, 1998–2022.

Supplemental Information 6 Prior sensitivity analysis of the June-cohort daily juvenile survival rate for northern bobwhites (Colinus virginianus) in southern Georgia, USA, 1998–2022.

Results show the effect of prior specification on the posterior distribution for the juvenile survival parameter estimated via an integrated population model. The prior distribution was specified as either fully vague (black), using on an informative mean and standard deviation derived from Terhune, Chandler & Martin, 2017 (orange), and using the same informative mean but varying the standard deviation up (blue) or down (green) by 50%.

Supplemental Information 7 Prior sensitivity analysis of the July-cohort daily juvenile survival rate for northern bobwhites (Colinus virginianus) in southern Georgia, USA, 1998–2022.

Results show the effect of prior specification on the posterior distribution for the juvenile survival parameter estimated via an integrated population model. The prior distribution was specified as either fully vague (black), using on an informative mean and standard deviation derived from Terhune, Chandler & Martin, 2017 (orange), and using the same informative mean but varying the standard deviation up (blue) or down (green) by 50%.

Supplemental Information 8 Prior sensitivity analysis of the August-cohort daily juvenile survival rate for northern bobwhites (Colinus virginianus) in southern Georgia, USA, 1998–2022.

Results show the effect of prior specification on the posterior distribution for the juvenile survival parameter estimated via an integrated population model. The prior distribution was specified as either fully vague (black), using on an informative mean and standard deviation derived from Terhune, Chandler & Martin, 2017 (orange), and using the same informative mean but varying the standard deviation up (blue) or down (green) by 50%.

Supplemental Information 9 Prior sensitivity analysis of the September-cohort daily juvenile survival rate for northern bobwhites (Colinus virginianus) in southern Georgia, USA, 1998–2022.

Results show the effect of prior specification on the posterior distribution for the juvenile survival parameter estimated via an integrated population model. The prior distribution was specified as either fully vague (black), using on an informative mean and standard deviation derived from Terhune, Chandler & Martin, 2017 (orange), and using the same informative mean but varying the standard deviation up (blue) or down (green) by 50%.

Supplemental Information 10 Effect of juvenile daily survival prior specification on density (birds/ha) of northern bobwhites (Colinus virginianus) in southern Georgia, USA, 1998–2022.

The prior distribution was specified as either fully vague (black), using on an informative mean and standard deviation derived from Terhune, Chandler & Martin, 2017 (orange), and using the same informative mean but varying the standard deviation up (blue) or down (green) by 50% in an integrated population model.

Supplemental Information 11 Prior (black) and posterior (orange) distributions of June-cohort juvenile daily survival rates of northern bobwhites (Colinus virginianus) in southern Georgia, USA, 1998–2022.

The prior distribution was specified using an informative mean and standard deviation derived from Terhune, Chandler & Martin, 2017.

Supplemental Information 12 Prior (black) and posterior (orange) distributions of July-cohort juvenile daily survival rates of northern bobwhites (Colinus virginianus) in southern Georgia, USA, 1998–2022.

The prior distribution was specified using an informative mean and standard deviation derived from Terhune, Chandler & Martin, 2017.

Supplemental Information 13 Prior (black) and posterior (orange) distributions of August-cohort juvenile daily survival rates of northern bobwhites (Colinus virginianus) in southern Georgia, USA, 1998–2022.

The prior distribution was specified using an informative mean and standard deviation derived from Terhune, Chandler & Martin, 2017.

Supplemental Information 14 Prior (black) and posterior (orange) distributions of September-cohort juvenile daily survival rates of northern bobwhites (Colinus virginianus) in southern Georgia, USA, 1998–2022.

The prior distribution was specified using an informative mean and standard deviation derived from Terhune, Chandler & Martin, 2017.

Supplemental Information 15 Correlation between annual female reproduction parameters and model estimates of April density for a population of Northern bobwhites (Colinus virginianus) in southern Georgia, USA, 1998–2022.

Correlations were generated between posterior samples of April density derived from an integrated population model and the observed percentage of radiotracked females surviving the breeding season which initiated nests (A), the observed percentage of nesting females surviving the breeding season which initiated renest attempts after a failed first brood (B), or the observed percentage of nesting females surviving the breeding season which initiated double-brood attempts after a successful first brood (C). Points represent the mean of posterior samples for April density (x-axis) and the mean of observed reproduction parameters (y-axis). Horizontal bars represent the 95% credible intervals of posterior samples for April density. Mean correlation coefficients are shown, as well as 95% credible intervals in parentheses.

Supplemental Information 16 Correlation between annual model estimates of demographic parameters and spring population growth rates for a population of Northern bobwhites (Colinus virginianus) in southern Georgia, USA, 1998–2022.

Correlations were generated between posterior samples derived from an integrated population model. Points represent the mean of posterior samples for population growth rates and demographic parameters. Horizontal and vertical bars represent the 95% credible intervals of posterior samples for demographic rates and population growth rates, respectively. Mean correlation coefficients are shown, as well as 95% credible intervals in parentheses. Dashed horizontal lines represent no population growth (lambda=0).

We thank the many technicians who helped collect the demographic data.

Additional Information and Declarations

Competing Interests

Author Contributions

Animal Ethics

Data Availability

The authors declare that they have no competing interests.

William B. Lewis analyzed the data, prepared figures and/or tables, authored or reviewed drafts of the article, and approved the final draft.

Chloé R. Nater conceived and designed the experiments, authored or reviewed drafts of the article, and approved the final draft.

Justin A. Rectenwald performed the experiments, authored or reviewed drafts of the article, and approved the final draft.

D. Clay Sisson performed the experiments, authored or reviewed drafts of the article, and approved the final draft.

James A. Martin conceived and designed the experiments, analyzed the data, authored or reviewed drafts of the article, and approved the final draft.

The following information was supplied relating to ethical approvals (i.e., approving body and any reference numbers):

Tall Timbers Research Station provided full approval for this research.

The following information was supplied regarding data availability:

The data and R and NIMBLE code for running the IPM are available at GitHub and Zenodo:

- https://github.com/wblewis7/S.GA.NOBO.IPM/tree/v1.2

- wblewis7. (2024). wblewis7/S.GA.NOBO.IPM: Publishing Data and Code for Lewis et al. NOBO IPM Manuscript (v1.2). Zenodo. https://doi.org/10.5281/zenodo.12825016.

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
