# Peer review of "Use of integrated population models for assessing density-dependence and juvenile survival in Northern Bobwhites (Colinus virginianus)"

_PeerJ, doi:10.7717/peerj.18625_

## Round 0.1 · original submission · Minor Revisions

Dear author,

Thank you for your submission. We have received two strong reviews and I agree with the reviewers that your work will be a valuable addition to the literature, pending some minor adjustments.

Best wishes,
Anthony

·

Basic reporting

This paper is well written. It includes sufficient and appropriate refences. Although there were ample supplemental figures and explanations provided, I did not see links to data or code, but I presume/hope the authors will provide those at a later date.

Experimental design

I have no concerns about experimental design. The study is especially impressive for having maintained consistent and rigorous data collection for so many years in a row.

Validity of the findings

There was nothing to make me question the validity of the findings.

Additional comments

This is a well-written and well-analyzed paper based on an amazing long-term data set. Kudos for a job well done.
1) Line 339 (equation for effects of abundance on non-breeding survival): I have no concerns with how you did this, summing across all individuals and estimating a common parameter for density dependence (gamma) over all ages, sexes, and years). But did you consider trying to estimate gamma separately for ages or sexes (in particular, are young birds more susceptible to DD, or maybe adult females due to rigors of reproductive investment?). It would also be neat to explore whether all birds in the population are equal in terms of their density-dependent influence on vital rates (thinking of Marlene Gamelon’s work on Great and Blue tits). What if instead of summing over adult males, adult females, and combined-sex subadults, N was a vector of these 3 population subcomponents, you have a dcat vector of 3 proportions that sum to 1, and you have gamma. The density dependence effect is gamma * dcat[1:3] * N[1:3] and if the estimated dcat isn’t c(0.33, 0.33, 0.33) you see which cohort has the greatest influence on the other cohorts.
2) Line 381: I consider “censoring independent of fate” to be one of the most important assumptions in known-fate analysis. If predators destroy transmitters at the same time they kill birds, it can lead to underestimates of mortality. Can you comment briefly (perhaps supplementally) on how censoring decisions were made and addressed? Or cite other studies and what they did with respect to censoring.
3) Paragraph 617-644: Somewhere in this paragraph I’d like to see some acknowledgement that whenever you have a completely (or mostly) latent vital rate, there is the danger that it will be biased or contaminated by other unmeasured variables, biases in other parameters, or model misspecification (e.g. getting nesting effort wrong). You don’t need to overdo it, but citing Riecke et al and/or Paquet et al would be appropriate here.
Minor Points:
4) Line 52: Change “harvest rates” to “harvest regulations”? Managers can’t fine tune harvest rates, they can only fiddle with regulations to incentivize or disincentivize participation and success. Perhaps your managed private hunting area was different, but in the waterfowl world we refer to this as “partial controllability” of harvest rates.
5) Line 57-58: Brook & Bradshaw 2008 and Knape & de Valpine 2012 are a couple of gems that also include analysis of numerous empirical datasets.
6) Line 84: One notable shortcoming of including DD in an IPM is that the inevitable circularity can lead to very slow-running models. You might include that as one shortcoming of the approach.
7) Line 87: Change process to processes.
8) Line 114: All of this is true, but maybe interesting and fun to point out that Errington’s early studies of Bobwhite in Wisconsin (although wildly speculative) helped shape some of the early ideas about density dependence and population regulation?
9) Line 124-125: Mention these alternative methods explicitly, e.g. attaching transmitters to tiny chicks and hoping they don’t influence survival. Or maybe just include citations to papers that measured survival by marking chicks (if any have) or by marking brood-tending adults with radios.
10) Lines 142-150: Sentences 2 and 4 are very redundant. Maybe combine sentences 1 & 2 to state “Our study…in Baker County in the Upper Coastal Plain physiographic region of southwest Georgia.” Omitted portions appear in sentence 4.
11) Lines 172-173: This sentence makes me wonder if you assumed a 50:50 sex ratio, and you did, but I don’t find out until much later. End this sentence by saying “so we assumed a 50:50 sex ratio among subadults”?
12) Line 183: Maybe not that important for this analysis, but I presume band reporting rates were also close to 100%.
13) Lines 227-229: This is not what independence means. If the probability of being including in the tracking data set is P1 and the probability of being in the productivity data set is P2, then P1*P2 individuals should be in both data sets, and if there is no overlap despite P1 and P2 both being reasonably large, then the data are not independent. Everybody repeats this “assumption” of IPMs and it is statistical/mathematical nonsense.
14) Line 324: Change on to of?
15) Line 332: Should this equation begin N.S rather than S?
16) Line 514: Lead is grammatically correct, but led would be less confusing.
17) Line 594: Clearer if you said “avoid basing management decisions off of erroneously identified drivers of population growth rates”.
18) Line 618-621: Not surprising given these were daily survival rates and you were taking the product of 100+ of them (i.e. Matthew Schofield’s ISEC plenary lecture). In cases like this, seemingly vague priors (e.g. Unif(0,1) are actually highly biased and lead to gross underestimates of cumulative survival).
19) Line 672: Roberts et al, 2024, J Fish & Wildlife Management have a new paper where they thoroughly explored within season capture probabilities for mallards and wood ducks.
20) Hixon et al. appears twice in Lit Cited.
21) Figs. 2-5: 95% credible intervals create the illusion that uncertainty is uniform along the entire credible interval, rather than being concentrated near the median. These might be too busy if you use violin plots, but I like seeing full posteriors when I can.
22) Fig 7: I assume you tried the classic version with upward and downward contributions to population change and thought this one was more informative? Consider adding color to make patterns jump out even more.
Todd W Arnold

·

Basic reporting

This manuscript is very clearly and cleanly written throughout. The methods and technical results may be challenging to readers who are not familiar with Bayesian methods, particularly integrated population models. Nevertheless, the methods and results are presented at a level of detail that is necessary for evaluation and replication of the work.
Professional English language is clearly used throughout with just a few grammatical errors may be word choice alternatives or sentence structuring alternatives that I point out below:
Line 20: the work among is preferred to between in reference to more than two elements.
Line 101: The phrase “with the difference inferring to inferring rates of juvenile survival” initially struck me as a bit unclear and I had to pause to think and understand it. I think the authors are referring differences between breeding productivity (e.g. hatched eggs/female) and post-breeding age ratios). I think this is explained more clearly later in the methods section but the sentence which spans lines 98-101 might be better separated into 2-3 sentences that more clearly convey what the authors are saying without getting too far into the details for the Introduction.
Line 143: “with” should be “within”
Line 166: “correspond” should be past-tense, as the methods and results should be written in past-tense. Also, try to make it more clear how the trapping periods trapping periods corresponded to breeding, post-breeding, and non-breeding seasons.
Line 216-220: Please explain how re-nesting attempts and double-brooding (I am not referring to male incubated nests) were handled if any hens incubated and more than one clutch. Were all productivity measures from nests of radio-tagged females expressed as no. chicks produced were from multiple nest attempts?
Line 236: Were the grids spread across all habitats or confined only to brood habitat types? There may be no need for this distinction if bobwhites did not move broods to habitats that were distinct vegetation types or locations separated from non-breeding or nesting habitat types.
Line 247 implies that the cells were representative of the entire study area, implying that there was no spatial segregation of broods across the study area compared to areas used during nesting and non-breeding. Please clarify.
Lines 258-260. Are there data or publications to cite which show there was no immigration of emigration from the study area. That you used radio-tagged birds suggest there are data that would support the claim that changes in abundance were limited to changes caused by variation in fecundity and survival.
Line 378: I think you meant to say that you square-rooted “bi-“monthly survival rates. I am not sure that “square-rooted” is a word or is used as a verb?
Line 431: Parenthetically list the parameters for which you used vague priors.
The background, context, and literature cited clearly explain the importance and relevance of this work to population ecology, integrated population models, and the study species are clearly explained, and well referenced in the introduction and those points are revisited in the discussion. Statements are well supported with references to relevant literature and results from the IPM analyses.
I commend the authors for presenting a useful summary of vital rates and population estimated parameters throughout the Population Demography section. The summary is logically organized and concisely summarized so that readers can easily glean vital rate parameters that are of interest to managers and scientists that study this species.
The following section titled Demographic Drivers of Population Growth are a bit more technical but clearly and concisely convey important findings.
The tables and figures are all relevant, of publication quality, and clearly labeled and described with fully informative captions. The raw data are not supplied, presumably because the IPM was based on complex, voluminous and varied data sets that would be too cumbersome or impossible to include. However, informative and vague priors for vital rates and other model parameters are included.

Experimental design

This study represents a synthesis of long term (25 year) population data sets toward well-defined ends to evaluate the response of a bobwhite to management and the relative effect of population vital rates (i.e. seasonal survival and productivity) on population dynamics of northern bobwhites, clearly within the scope of PeerJ.
The research questions and application to wildlife conservation problems are clearly defined and highly relevant to understanding how bobwhite populations specifically, and wildlife populations in general respond to management and environmental change. This study represents a rigorous application of integrated population models to better understand how populations of a species of high conservation concern function and respond to management applied at local and regional scales.
Modeling the dynamics of wildlife populations with empirical data is challenging because certain parameters are difficult to estimate, especially post-breeding survival of juveniles after hatching. The authors estimated this parameter from empirical data in a novel way, by aging young of year captured or harvested during the fall to account for survival of cohorts hatched over a 4-5 month period. Thus, an important parameter that is difficult to measure empirically was estimated in a unique way that could have only been done with an IPM. The authors demonstrate how their estimate of post-hatching survival was improved with informed priors based on their method. The authors were also innovative in their treatment of abundance data which are essential to IPM, accounting for process and measurement uncertainties in covey density and cover size data collected from census grids. Taken together an important contribution of the research is a rigorous evaluation of the effect of density dependence on population growth which has not received sufficient attention in other population models for this species due to lack of empirical data. An important contribution of this research is a rigorous evaluation of density dependence in this species, something that was especially important for a study population that was likely near the limit of environmental carrying capacity.
Overall, this was a highly rigorous investigation of population dynamics using long-term and complex empirical data sets analyzed with state-of-art technical methods, objectively and ethically presented to answer fundamental scientific and management questions.
The methods and sources of data are described with sufficient detail that others proficient with integrated population models could replicate the work performed here with independent data sets. The presentation of methods, though complex, is on-par with other IPM studies reported in the scientific literature.

Validity of the findings

This study and manuscript are based on large, robust and integrated data sets, ideally suited to Integrated population modeling and thoroughly described in the methods section. This is one of a few integrated population or several other types of models that have been published for different bobwhite populations. To my knowledge the model presented here is based on the most robust data set available for this species and demonstrates quite well the application of IPMs to scientific questions and population management problems.
The conclusions are overall clearly stated with comprehensive reference to relevant original research. The conclusions are clearly supported by results and do not go beyond what could be supported by the data and modeling results.
I do not see any areas where this article fails to meet the standards of PeerJ.

Additional comments

The discussion thoroughly and concisely presents relevant and well-supported interpretation of the results with reference to key literature and results of preceding studies. I was particularly pleased to see the authors point out that their study population was intensively managed with food supplementation, mammalian predator control, and management of forest stand density during most of the years in their 25-year data set.
The authors also appropriately recognize that they studied a high-density and stable to increasing population which inhabits an environment that is nearly optimal for the species. The study population is also at the southern end of the species’ range where breeding productivity is known to have a stronger effect on population dynamics compared to the effect of non-breeding survival in more northern areas where populations are in decline.
The study population was also lightly hunted (4% of mortality was attributed to harvest) unlike many other northern bobwhite populations. Based on results from other population models the authors correctly acknowledge that there is regional variation in abundance and dominant vital rates and degrees or mechanisms of density dependence that might produce different results and conclusions from the results of this study. The results from this study could be more strongly placed in the context of population models and population dynamics studies conducted in other regions where population trajectories and environmental conditions differ.
It would be interesting to compare the dominance of vital rates and degree of density dependence among different segments of the annual time series when the population was increasing (1998-2002) declining (2003-2010) and stable (2012-2022). Such comparisons if feasible would add a nice dimension to this manuscript, particularly because these time spans also seem to correspond with different management regimes or environmental conditions.
This is an outstanding piece of research that is worthy of publication in PeerJ with minimal revisions of the manuscript. This work stands out in my mind of representing the state of art of application of IPMs in addressing questions about population dynamics of resident game birds and other species of conservation concern.
My primary suggestions to the authors are to provide more of a synthesis of findings from other IPMs of density dependence effects and dominance of vital rates in high- versus low- density or increasing versus declining populations of this species. The authors started down this road but could emphasize these comparisons more. In short, I would encourage the authors to broaden the scope of the Discussion from this particular IPM to what the results of IPMs and other modeling efforts have shown across the range of geography and environments inhabited by northern bobwhites.
An especially useful addition to the manuscript would be to compare density dependence and posterior parameter estimates of vital rates for the three periods I identified in the authors’ 25-year data set (see above).

---

## Round 0.2 · accepted · Accept

Dear author,

Thank you for your resubmission. I have reviewed the updated manuscript and am happy that your edits constitute appropriate responses to the reviewers' comments. Similarly, I agree with your position where you have disgareed with the reviewers. I am therefore pleased to recommend acceptance and publication.

With regards to your query about data availability, I can confirm that the Data Availability statement will be placed at the end of the published manuscript, hence will be available to readers.

Best wishes,
Anthony